# Detection of ASD Children through Deep-Learning Application of fMRI

**DOI:** 10.3390/children10101654

**Published:** 2023-10-05

**Authors:** Min Feng, Juncai Xu

**Affiliations:** 1Nanjing Rehabilitation Medical Center, The Affiliated Brain Hospital, Nanjing Medical University, Nanjing 210029, China; 2School of Chinese Language and Literature, Nanjing Normal University, Nanjing 210024, China; 3School of Engineering, Case Western Reserve University, Cleveland, OH 44106, USA; juncai.xu@gmail.com

**Keywords:** autism spectrum disorder, ASD screening, convolutional neural networks, deep learning, fMRI

## Abstract

Autism spectrum disorder (ASD) necessitates prompt diagnostic scrutiny to enable immediate, targeted interventions. This study unveils an advanced convolutional-neural-network (CNN) algorithm that was meticulously engineered to examine resting-state functional magnetic resonance imaging (fMRI) for early ASD detection in pediatric cohorts. The CNN architecture amalgamates convolutional, pooling, batch-normalization, dropout, and fully connected layers, optimized for high-dimensional data interpretation. Rigorous preprocessing yielded 22,176 two-dimensional echo planar samples from 126 subjects (56 ASD, 70 controls) who were sourced from the Autism Brain Imaging Data Exchange (ABIDE I) repository. The model, trained on 17,740 samples across 50 epochs, demonstrated unparalleled diagnostic metrics—accuracy of 99.39%, recall of 98.80%, precision of 99.85%, and an F1 score of 99.32%—and thereby eclipsed extant computational methodologies. Feature map analyses substantiated the model’s hierarchical feature extraction capabilities. This research elucidates a deep learning framework for computer-assisted ASD screening via fMRI, with transformative implications for early diagnosis and intervention.

## 1. Introduction

Autism spectrum disorder (ASD) is a complex neurodevelopmental disorder that is characterized not only by impairments in social interaction and communication but also by restricted, repetitive patterns of behavior, interests, or activities, as outlined in DSM-5 criteria [1,2]. The imperative for early diagnostic endeavors is crucial for the initiation of targeted therapeutic interventions. However, traditional diagnostic paradigms hinge largely on subjective evaluations, often resulting in protracted and indeterminate diagnostic outcomes [3,4]. In contrast, functional magnetic resonance imaging (fMRI) has emerged as an invaluable instrument for the objective and meticulous diagnosis of ASD [5,6,7,8]. Beyond its diagnostic utility, fMRI is pertinent in personalized metabolic therapies in ASD, such as integrated PET/MRI systems, and offers the advantage of no exposure to radiation. To further substantiate the utility of fMRI in ASD diagnostics, it is worth noting that fMRI studies have revealed distinct neural connectivity patterns in children with and without ASD [9]. Specifically, increased local connectivity and decreased long-range connectivity have been observed in ASD subjects [10]. These findings have guided the selection of fMRI variables that were integrated into our deep-learning algorithm.

Building on these earlier observations, numerous scholarly investigations have rigorously probed the applicability of fMRI in the ASD diagnostic landscape, frequently incorporating machine-learning frameworks to augment diagnostic precision [11,12,13,14,15,16]. Foundational research, for instance, scrutinized fMRI’s capacity to discern aberrant neural connectivity in individuals afflicted with ASD, while supplementary inquiries concentrated on algorithmic automation [9,17,18]. Subsequent scholarship delved into the integration of machine-learning algorithms within the analytical matrix of fMRI data, the elucidation of latent neural architectures that are pertinent to ASD, and the implementation of complex computational models for diagnostic purposes [17,19,20,21]. Despite these advancements, the scholarly focus remains disproportionately oriented toward adult populations, thereby creating a conspicuous void in pediatric-focused research [19,22,23]. A review of prior deep-learning studies reveals a focus on adult populations, the employment of various fMRI data, and the involvement of children with different characteristics.

Convolutional neural networks (CNNs) have garnered substantial acclaim for their proficiency in image-recognition tasks that have extended into the domain of medical imaging [24,25,26]. The intrinsic capability of CNNs to autonomously and adaptively decipher spatial feature hierarchies renders them uniquely apt for the intricate analysis of fMRI datasets [27,28,29]. Despite the availability of the Autism Brain Imaging Data Exchange (ABIDE I) dataset since 2014 and the NDAR dataset since 2006, there exists a significant lacuna in scholarly literature that integrates fMRI data with CNN methodologies for the targeted early diagnosis of ASD in pediatric populations.

The overarching objective of the current investigation was to address these existing research gaps. A CNN framework was deployed for the rigorous analysis of fMRI scans, targeting the incipient detection of ASD in pediatric subjects. In doing so, the study furnished a swift, unerring, and impartial diagnostic conduit, thereby enhancing the prospects for timely and effective therapeutic interventions. The seminal contributions of this study are delineated as follows:This study introduced a custom deep CNN for fMRI data, achieving high accuracy in distinguishing pediatric ASD from typical development.The study uncovered the model’s feature learning through in-depth analysis and feature-map visualization, enabling robust ASD discrimination.The investigation yielded invaluable insights via feature-map analysis, corroborating the efficacy of the CNN architecture in hierarchical feature learning.

This study employed convolutional neural networks to scrutinize fMRI data for the nascent diagnosis of ASD in pediatric demographics. Subsequent sections of this article elucidate the data acquisition, preprocessing, and segmentation protocols that were used to compile an exhaustive dataset (Section 2); the proposed CNN architecture and its training regimen (Section 3); the empirical evaluation encompassing performance metrics, confusion matrix analytics, and feature-map visualizations (Section 4); and the concluding remarks and avenues for future research (Section 5).

## 2. Materials and Methodology

### 2.1. Data Collection and Preprocessing

MRI scans, obtained with a 3 Tesla Allegra, were performed within three months post-diagnosis for pediatric subjects. The data for this study were obtained from the Autism Brain Imaging Data Exchange (ABIDE I) repository, specifically originating from the New York University Langone Medical Center [17]. This subset encompassed 56 individuals who were clinically diagnosed with ASD and 70 age-matched controls who exhibited typical development (TD). Children who were on psychostimulants ceased medication 24 h before scanning, pending physician approval. During the resting-state fMRI, most participants relaxed with open eyes, focusing on a white cross-hair displayed against a black screen. The phenotypic attributes of the participants from the NYU dataset are systematically detailed in Table 1.

In our rigorous investigation, we utilized pediatric fMRI data that had undergone a meticulous preprocessing regimen via the connectome computation system (CCS). This widely accessible computational framework offered an exhaustive preprocessing pipeline. Specific procedures within this pipeline encompassed a range of techniques, from the registration of anatomical brain masks to functional images using FMRIB’s linear image registration tool (FLIRT) to slice-timing correction achieved with 3dTshift. Additional steps included skull stripping via Analysis of Functional NeuroImages’ 3dAutomask, motion correction through 3dvolreg, voxel intensity normalization, nuisance signal removal, and band-pass filtering in the 0.01–0.1 Hz frequency range.

In the data-preprocessing phase, the four-dimensional fMRI datasets, stored in the Neuroimaging Informatics Technology Initiative (NIFTI) format, underwent a series of complex transformations. The initial operation entailed the decomposition of each four-dimensional dataset into a series of three-dimensional volumes, each corresponding to a specific temporal instance (Figure 1).

Mathematically, this decomposition is articulated as follows (Equation (1)):(1)$$V(x,y,z)=F(x,y,z,t_{i})$$
where V(x,y,z) represents the three-dimensional volume extracted at a particular time $t_{i}$.$F(x,y,z,t_{i})$ denotes the original four-dimensional fMRI dataset, and $t_{i}$ specifies the temporal instance under consideration.

Subsequent to this, the extracted three-dimensional volumes were further partitioned into two-dimensional slices. These slices were oriented along the anatomical planes—sagittal, coronal, and axial—and were amalgamated to form a two-dimensional, three-channel data representation.

The mathematical formalism for this partitioning is encapsulated in the following equation (Equation (2)):(2)D=[S(x,z),C(y,z),A(x,y)]
where D constitutes the two-dimensional, three-channel data representation. Here, S(x,z), C(y,z), and A(x,y) denote slices obtained along the sagittal, coronal, and axial planes, respectively. Specifically, $\;S\left(x,\;z\right)=V\left(x,\;y\_i,\;z\right)$, $C\left(y,\;z\right)=V\left(x\_i,\;y,\;z\right)$, and $A\left(x,\;y\right)=V\left(x,\;y,\;z\_i\right)$.

This intricate transformational sequence culminated in the conversion of each four-dimensional fMRI dataset into an array comprising 176 two-dimensional images. This array served as a rich substrate for subsequent analytical investigations.

This methodological approach generated a voluminous dataset of 2D echo planar data in three channels, amounting to 22,176 samples. Of this corpus, 17,740 samples were earmarked for the training set, while an additional 20% were set aside for validation purposes. The remaining 4436 samples constituted the test dataset. This meticulous procedure of data transformation and slicing effectively facilitated the assembly of a comprehensive echo planar dataset, optimally configured for subsequent deep-learning analyses.

### 2.2. Convolutional Neural Network Framework

In the present study, the CNN model was employed to analyze 2D echo planar data obtained from fMRI scans for the purpose of recognizing ASD in pediatric populations. The architectural framework of the model incorporated convolutional layers, max pooling layers, dense layers, batch normalization, the dropout technique, and a terminal softmax activation function [30,31,32,33]. To facilitate binary classification, the cross-entropy loss function was utilized, while model optimization was achieved through the Adam optimizer [34,35].

The convolutional layer is fundamental to any CNN. It applies filters to the input image to create feature maps, capturing local patterns [30,36]. The equation for this operation is as follows:(3)$$O_{ij}=\sum_{m}\sum_{n}D_{\left(i+m)(j+n\right)}\times K_{mn}$$
where $O_{ij}$ is the output feature map, D is the input image, K is the kernel, and m, n are the dimensions of the kernel.

Max pooling layers serve to downsample the spatial dimensions of the input volume, thereby emphasizing salient features for more efficient computational processing. The equation for max pooling is as follows:(4)M=max(R)
where M is the max-pooled output and R is a region in the input feature map.

Dense layers perform complex transformations on the flattened feature maps, contributing to the final classification. The equation for the dense layer is as follows:(5)y=Activation(W×x+b)
where y denotes the output, W are the weights, x is the input, and b is the bias term.

Batch normalization normalizes the output from the activation function, aiding in faster and more stable training. The equation for batch normalization is as follows [32]:(6)$$\hat{x}=\frac{x-\mu }{\sqrt{\sigma^{2}+\epsilon }}$$
where $\hat{x}$ is the normalized output, x is the input, μ is the mean, $\sigma^{2}$ is the variance, and ε is a small constant introduced to prevent division by zero.

The softmax activation function converts the network’s output into a probability distribution over the target classes. The equation for softmax is as follows:(7)$$S(y_{i})=\frac{e^{y_{i}}}{\sum_{j}e^{y_{j}}}$$
where $S(y_{i})$ is the softmax output for class i, $y_{i}$ is the input for class i, and $y_{j}$ is the input for all classes.

For the binary classification task, the cross-entropy loss function serves as the chosen loss metric. The equation for this loss function is as follows [34]:(8)$$L(y,\hat{y})=-(y\log(\hat{y})+(1-y)\log(1-\hat{y}))$$
where L is the loss, y is the actual label, and $\hat{y}$ is the predicted label.

To optimize the model, the Adam optimizer was employed. The equation for the Adam optimizer is as follows [35]:(9)$$\theta_{t+1}=\theta_{t}-\frac{\alpha \cdot m_{t}}{\sqrt{v_{t}}+\epsilon }$$
where θ  are the parameters, α is the learning rate, $m_{t}$ and $\;v_{t}$ are the first and second moment estimates, respectively, and ϵ is a small constant to avoid division by zero.

CNNs undergo parameter optimization via backpropagation algorithms that are aimed at minimizing a predefined loss function [37]. The optimizer updated the weights based on the gradients computed during backpropagation, refining the model for better performance.

### 2.3. Workflow for ASD Recognition Using CNNs

The establishment, training, and testing of a CNN model for ASD recognition is a rigorous, multi-step process that adheres to academic standards and mathematical formalism. The process commenced with the acquisition of 4D fMRI data from the ABIDE dataset (Algorithm 1). These data underwent comprehensive preprocessing, including motion correction and normalization, and were then decomposed into 3D volumes and 2D anatomical slices. These slices served as the CNN input, facilitating subsequent model initialization, training, and testing.
**Algorithm 1:** CNN Model for ASD Recognition**Data:** Preprocessed 4D fMRI data *F*(*x,y,z,t*)
**Result:** ASD Recognition Performance
**1 Step 1: Data Preprocessing and Slicing**
**2 for**
*t_i_ in T (Temporal Instances)*
**do**
**3**    *V* (*x,y,z*) = *F*(*x,y,z,t_i_*);
**4**    *D* = [*S*(*x,z*)*,C*(*y,z*)*,A*(*x,y*)];
**5**  Add *D* to Dataset
**6  Step 2: Dataset Partitioning**
**7**  Divide Dataset into Training, Validation, and Test Sets
**8  Step 3: Model Initialization**
**9**  Initialize CNN Model Parameters *θ*
**10 Step 4: Model Training**
**for**
*i* = 1 *to N (Iterations)*
**do**
**11**   Train CNN Model on Training Set using *θ* and Adam Optimizer;
**12**   Validate Model on Validation Set **if**
*Validation Loss* ≤ *Threshold*
**13   then**
**14**     Break
**15 Step 5: Model Testing**
**16** Test Trained Model on Test Set
**17 Step 6: Performance Evaluation**
**18** Compute ASD Recognition Performance Metrics

In accordance with the aforementioned pseudocode, the procedural framework for ASD recognition is delineated as follows:

Step 1: Data Preprocessing and Slicing

4D fMRI datasets are converted into 3D volumes, each representing a specific time $t_{i}$, as $V\left(x,\;y,\;z\right)=F\left(x,\;y,\;z,\;t\_i\right)$. These volumes are sliced into 2D images along anatomical planes to form a 2D, three-channel data $D=\left[S\left(x,\;z\right),\;C\left(y,\;z\right),\;A\left(x,\;y\right)\right]$.

Step 2: Dataset Partitioning

The dataset is methodically partitioned into training, validation, and testing subsets, serving the distinct functions of model training, hyperparameter optimization, and evaluative performance assessment, respectively.

Step 3: Model Initialization

The CNN model is instantiated with an initial set of stochastic parameters θ, thereby establishing the commencement point for subsequent optimization.

Step 4: Model Training

The model is subjected to a series of iterative training cycles, employing the Adam optimization algorithm to fine-tune parameters and minimize the objective loss function. The training phase culminates either upon achieving a pre-specified validation loss threshold or after a predetermined epoch count.

Step 5: Model Testing

The fully developed model undergoes stringent evaluation on the test subset to assess its capacity for generalization and its aptitude in accurately classifying ASD cases.

Step 6: Performance Evaluation

A comprehensive suite of evaluative metrics, encompassing accuracy, precision, recall, and F1 score, are computed to rigorously quantify the model’s capabilities in ASD recognition.

## 3. Experimental Setup

### 3.1. Parameters for CNN

The given model is a CNN designed for image classification, comprising several key layers (Figure 2). The Conv2D layers are essential for feature extraction, applying filters to the input image to detect patterns like edges and textures. The MaxPooling2D layers reduce the spatial dimensions, summarizing the features in each region and thereby reducing overfitting. Batch normalization layers are used to normalize the activations, improving training speed and stability. The fully connected layers, commonly referred to as dense layers, execute classification tasks predicated on the features ascertained through prior computational layers, with the final dense layer using a softmax activation function to classify the input into two categories. Dropout layers are strategically integrated within the network architecture, serving to stochastically nullify a subset of input units during the training phase as a regularization technique to mitigate overfitting. Together, these layers form a robust architecture that is capable of learning complex patterns in the input data, making it suitable for various image-recognition tasks. The layer parameters of the proposed neural network are shown in Table 2.

The CNN model is tailored for binary classification, with a structure designed to balance complexity and computational efficiency. The “data shape” parameter, derived from the training data, ensures that the model aligns with the input dimensions, while the “kernel_size” of (3, 3) allows for the capture of local patterns without excessive computational cost. Initiating with 32 filters, the architecture incrementally doubles the quantity of convolutional kernels subsequent to each max-pooling layer, thereby augmenting its capacity for intricate pattern recognition. With “n_classes” set to 2, the output layer is configured for binary classification, using a softmax activation function to provide probability estimates. Collectively, these settings reflect a common and effective approach in deep learning, aimed at learning hierarchical features from image data for binary classification tasks.

### 3.2. Training Model

The algorithm designed for ASD recognition was instantiated on a Python 3 environment and architected atop the Keras framework. For enhanced computational efficiency and expedited training, the model was deployed on an NVIDIA A100 graphics processing unit (GPU). This amalgamation of hardware and software specifications engendered both robust performance and scalability, thereby facilitating the seamless execution of the algorithm’s intricate mathematical computations and iterative training cycles.

For the model compilation, the loss function employed was “crossentropy”, which is particularly apt for multi-class classification scenarios, where labels are represented as integers. The optimization of the model’s weights was conducted via the “Adam” optimizer, and the performance was gauged using the “accuracy” metric. The training process was delineated across 50 epochs with a batch size of 64, implying that weight adjustments occurred every 64 samples. A fraction of 20% of the training data was earmarked for validation, serving as an evaluative gauge for the model’s generalization capabilities. This orchestrated confluence of hyperparameters and methodologies constituted the training regimen, guiding the model to discern intricate patterns in the data for effective classification. Upon the culmination of this training regimen, the wall time—indicative of the actual elapsed duration for model training—was recorded at 1 min and 52 s. Figure 3 and Figure 4 elucidate the trajectory of accuracy and loss metrics throughout the training phase.

The training history data—comprising accuracy, validation accuracy, loss, and validation loss—were retrieved from a file and plotted over 50 epochs. Two line plots were generated: one for accuracy and the other for loss. In each plot, training and validation metrics were distinguished by separate lines and colors, with markers emphasizing values at each epoch. Labels, legends, and suitable font sizes were incorporated for clarity.

The accuracy plot revealed the model’s learning trajectory in precise classification. The sky-blue line for training accuracy and the coral line for validation accuracy presented the model’s efficacy on both datasets across 50 epochs. An ascending trend in both lines signified effective learning, while a notable discrepancy may have hinted at overfitting or underfitting. The loss plot delineated the decrement in the model’s error over time. Sky-blue for training loss and coral for validation loss represented errors on respective data. A declining trend in both was usually favorable, reflecting error minimization. Conversely, a rising validation loss, in contrast to a declining training loss, may have signified overfitting, a condition in which the model performed exceptionally well on the training data but exhibited poor generalization to new, unseen data.

### 3.3. Evaluation Metrics

In the assessment of binary classification models, four principal metrics—accuracy, precision, recall, and F1 score—are employed to gauge a model’s performance. These metrics collectively furnish a multifaceted evaluation of a model’s capability to accurately categorize instances. The mathematical formulations for these key metrics are as follows:(10)$$\mathrm{Accuracy}=\frac{\mathrm{TP}+\mathrm{TN}}{\mathrm{TP}+\mathrm{TN}+\mathrm{FP}+\mathrm{FN}}$$
(11)$$\mathrm{Precision}=\frac{\mathrm{TP}}{\mathrm{TP}+\mathrm{FP}}$$
(12)$$\mathrm{Recall}=\frac{\mathrm{TP}}{\mathrm{TP}+\mathrm{FN}}$$
(13)$$F1\text{-}\mathrm{Score}=\frac{2\times \mathrm{Precision}\times \mathrm{Recall}}{\mathrm{Precision}+\mathrm{Recall}}$$
where TPs is true positives, TNs are true negatives, FPs are false negatives and FNs are false negatives.

## 4. Results and Analysis

### 4.1. Model Performance

Utilizing these four indicators in tandem allows for a nuanced understanding of a model’s strengths and weaknesses, guiding further refinement and optimization. The four indicators for the proposed model are shown in Table 3.

The efficacy of the proposed model was quantified by multiple key metrics, as delineated in the accompanying table. An impressive accuracy rate of 99.39% underscored the model’s robust capability for accurate instance classification. The precision rate of 99.85% reflected the model’s accuracy in predicting positive instances, indicating that almost all predicted positives were true positives. The recall rate, at 98.80%, represented the model’s sensitivity in identifying all actual positive instances, while the F1 score of 99.32% offered a balanced view of precision and recall, providing a holistic metric of the model’s robustness. Collectively, these metrics highlighted the model’s exceptional performance in both general classification and the nuanced handling of positive instances.

The confusion matrix (Figure 5) served as a graphical elucidation of the model’s discriminative efficacy between the two diagnostic categories: “TD” and “ASD”. Within this matrix, rows delineated the ground-truth classifications, while columns signified the model’s predicted outcomes. The percentage values showed the proportion of correct and incorrect classifications: 99.88% of “TD” were correctly classified, and only 0.12% were misclassified as “ASD”; 98.80% of “ASD” were correctly identified, and 1.20% were misclassified as “TD”. These percentages highlighted the model’s high accuracy in distinguishing between the two classes, with very low rates of misclassification. This granular perspective facilitated a comprehensive evaluation of the model’s merits and potential avenues for enhancement, thereby substantiating its robust performance in the specified classification task.

To further validate the CNN model, a five-fold cross-validation approach was employed, offering a more robust assessment of model performance compared to a singular train/test split. This method entailed partitioning the data into five equal subsets, followed by five iterations of training and evaluating the model—each iteration utilizing a distinct subset as the test set while amalgamating the remaining data for training. Through the examination of model performance across these multiple folds, a more reliable estimation of the model’s generalization capacity on unseen data was attained.

The fold accuracies, as delineated in Table 4, showcased a commendable level of consistency, signifying proficient performance of the model across varied data subsets. These values denoted the proportion of correct classifications within each fold, hence elucidating the model’s robustness. Despite a high accuracy continuum across all folds, it accentuated the model’s adeptness in differentiating between the “TD” and “ASD” classes.

### 4.2. Feature Map Analysis

The given model architecture comprised a three-layer convolutional deep network that was designed to process fMRI graphics, and it provided a hierarchical approach to feature extraction (Figure 6, Figure 7 and Figure 8).

The inaugural Conv2D layer ingested fMRI images and employed a predetermined quantity of filters with a designated kernel size, subsequently invoking a ReLU activation function (Figure 6). This layer was instrumental in discerning low-level features, such as edges and rudimentary shapes, within the fMRI data. The subsequent BatchNormalization normalized the activations, and MaxPooling2D reduced the spatial dimensions, focusing on the most important information. The quantity of filters was subsequently doubled for the ensuing layer.

The next Conv2D layer, with twice as many filters of the first, captured more complex and abstract features, such as textures and patterns (Figure 7). The ReLU activation introduced non-linearity, allowing the model to learn more intricate representations. Subsequent to the convolutional layer, BatchNormalization and MaxPooling2D were employed, the latter serving to further downsample the feature maps. Concurrently, the quantity of filters was doubled.

The third Conv2D layer continued this process, now with four times the original number of filters (Figure 8). At this stage, the model was likely detecting high-level characteristics, such as specific regions or structures within the fMRI images, that were germane to the specific diagnostic task. The ReLU activation, BatchNormalization, and MaxPooling2D continued the pattern of introducing non-linearity, normalizing activations, and reducing spatial dimensions.

Throughout these three layers, the model transitioned from detecting simple, spatial features to more abstract and complex characteristics. The progressive doubling of filters and the consistent use of ReLU, BatchNormalization, and MaxPooling2D created a hierarchy of feature maps that increased in abstraction and complexity. This hierarchical structure was particularly suitable for interpreting fMRI graphics, where understanding both local details and global structures is essential. The final fully connected layers and dropout layers further refined the model’s ability to classify based on these extracted features.

In this experiment, 500 samples were randomly extracted from the test set to investigate the role of feature maps in a deep neural network across four layers—three convolutional and one fully connected (Figure 9). After applying t-SNE for dimensionality reduction, the feature maps from these layers were visualized as scatter plots. The first convolutional layer showed moderate feature extraction capabilities, capturing basic elements like edges and textures, but with less distinct clustering. As we progressed to the second and third convolutional layers, the clusters became increasingly defined, indicating the improved feature extraction of more complex structures. The fully connected layer’s feature map ideally displayed the most distinct clusters, indicating that it effectively combined high-level features for final classification.

The analysis of these feature maps leads to several conclusions. First, the network demonstrates a clear hierarchy of feature extraction, with each subsequent layer capturing increasingly complex features. This is evident from the progressively distinct clusters formed in the scatter plots. Second, the third convolutional stratum and the fully connected tier exhibit the most promising discriminatory power, suggesting that they play a critical role in effectively classifying between “TD” and “ASD”. These observations affirm the importance of deeper layers in the network for complex classification tasks, thereby validating the architecture’s efficacy.

### 4.3. Discussion

The proposed CNN model demonstrated a remarkable capability in distinguishing ASD subjects from TD subjects. This robust performance, substantiated by the model’s hierarchical feature extraction, suggested that the model captured meaningful and generalizable representations from the fMRI data. The high accuracy of the model has significant ramifications for the field of ASD diagnosis. Unlike traditional diagnostic methods, which often rely on subjective assessments and which are time-consuming, the proposed model offers a rapid, objective, and highly accurate diagnostic tool. This could be particularly transformative for preliminary ASD screening and could also contribute to the identification of novel biomarkers or neural signatures associated with ASD.

When the model’s performance is juxtaposed with existing literature, its superiority becomes evident (Table 5). This study introduced a CNN model that not only attained an unparalleled accuracy rate of 99.39% in differentiating ASD from typical development, but also provided expeditious, objective, and highly precise ASD screening, with transformative potential for early diagnosis and intervention. This accuracy substantially outperformed the closest competing study by Yakolli et al. (2023), which achieved 88.0% accuracy using a CNN model. The performance delta is even more pronounced when it is compared to other methodologies, such as deep-broad learning and decision models, which have accuracies around the 70% mark. The model’s exceptional performance was not confined to CNN-based methodologies, but extended to diverse computational approaches, including recurrent neural networks (RNNs) and capsule networks (CapsNets). This suggests that the unique architectural elements and feature-extraction strategies employed in the model rendered it exceptionally effective for the task at hand.

Expanding upon the model’s notable performance metrics, the data preparation phase involves segmenting extracted three-dimensional fMRI volumes into two-dimensional slices across three anatomical planes, thereby preserving maximal information from the original 3D dataset. The model incorporates 3 × 3 convolution kernels, ReLU activation functions, and MaxPooling layers to effectively capture spatial features at various scales. This specialized feature-extraction methodology addresses shortcomings in extant models, thereby contributing to the model’s enhanced performance. Furthermore, the model’s consistently high accuracy across a pediatric age range of 6–18 years and diverse ASD severity levels substantiates its generalizability and diagnostic sensitivity. The employment of whole-brain resting-state fMRI data, without regional isolation, accentuates the model’s capacity for identifying distributed neural patterns, thereby bolstering its robustness.

However, this study was not without limitations. The model’s efficacy was principally validated within a school-aged demographic, thereby raising questions regarding its generalizability to younger pediatric populations, which is essential for early ASD diagnosis. The acquisition of fMRI data from younger children with ASD presents challenges, due to sensory sensitivities and behavioral considerations. Consequently, the applications of alternative, child-friendly neuroimaging modalities, such as functional near-infrared spectroscopy (fNIRS) and electroencephalography (EEG), warrant further investigation. Additionally, the binary classification framework of the model fell short of capturing the nuanced heterogeneity of ASD symptoms and their varying degrees of severity. Moreover, eye-tracking methods, coupled with machine-learning techniques, offer another promising avenue for autism diagnosis, as exemplified by recent studies [45]. The binary classification nature of the model did not encapsulate the spectrum of ASD symptoms, which vary in severity and manifestation. The study also fell short in providing insights into the model’s decision-making rationale, which is a crucial factor for clinical adoption. Furthermore, the computational resources requisite for training the model may not be ubiquitously available, thereby constituting a potential impediment to its broad-scale deployment. Another area of critical concern is that ethical considerations are integral to the deployment of AI methodologies, such as CNNs, in ASD diagnosis. Such concerns encompass data confidentiality, informed consent, and the risk of algorithmic bias, which may disproportionately impact specific demographics. Addressing these ethical dimensions is vital for the responsible application and clinical acceptance of AI-based diagnostic systems.

## 5. Conclusions

This investigation pioneered the development of a CNN model for the incipient diagnosis of ASD in pediatric cohorts, utilizing fMRI datasets. The study encompassed meticulous data preprocessing, bespoke CNN architecture training, performance evaluation manifesting unparalleled accuracy metrics exceeding 99%, and feature-map visualization that substantiated the model’s adeptness in hierarchical feature learning. Collectively, the framework constituted a swift, objective, and remarkably precise computer-assisted diagnostic instrument for ASD screening, leveraging the synergies between fMRI data and advanced deep-learning algorithms.

Based on the investigation, the salient conclusions may be articulated as follows:(1)The proffered CNN model exhibited extraordinary proficiency in differentiating ASD from typically developing (TD) subjects, attaining an accuracy metric of 99.39%, thereby markedly eclipsing prior scholarly endeavors. This underscored the model’s efficacy in discerning discriminative features within the fMRI datasets.(2)Scrutiny of the model’s feature maps corroborated its capabilities in hierarchical feature extraction, with the more advanced layers serving a pivotal role in demarcating ASD from TD. This suggested that the model was adept at learning increasingly intricate and abstract data representations.(3)The model furnished a rapid, unerring, and highly precise diagnostic apparatus for ASD screening and identification, with the potential to revolutionize conventional, subjective, and protracted diagnostic frameworks.(4)The model’s exceptional performance across a gamut of computational methodologies accentuated the efficacy of its unique architectural design and feature extraction paradigms for this specific classification task.

This investigation elucidated a CNN model with an extraordinary accuracy rate exceeding 99% in the classification of ASD, utilizing fMRI datasets, thereby significantly outperforming extant computational methodologies. While this study’s findings are auspicious, the actualization of this technology’s clinical utility mandates further scholarly pursuits in model validation across heterogeneous demographic subsets, the integration of multi-modal data, the interpretability of algorithmic decisions, resource-efficient deployment, classification granularity across autism subtypes and severity gradations, and prospective real-world evaluations. Rigorous endeavors to surmount these limitations and translational obstacles will be indispensable for metamorphosing this high-performing deep-learning framework from a technological novelty into a clinically deployable decision-support mechanism, thereby enhancing early screening and personalized therapeutic interventions across the autism spectrum.

## Figures and Tables

**Figure 1 children-10-01654-f001:**
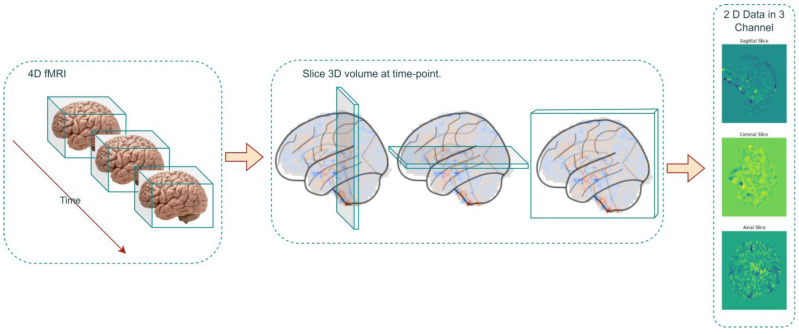
Process of converting 4D fMRI image to 2D echo planar data.

**Figure 2 children-10-01654-f002:**
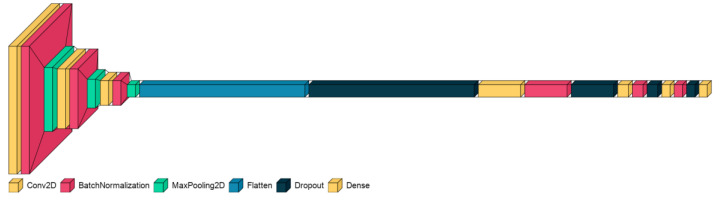
An advanced CNN framework for ASD classification.

**Figure 3 children-10-01654-f003:**
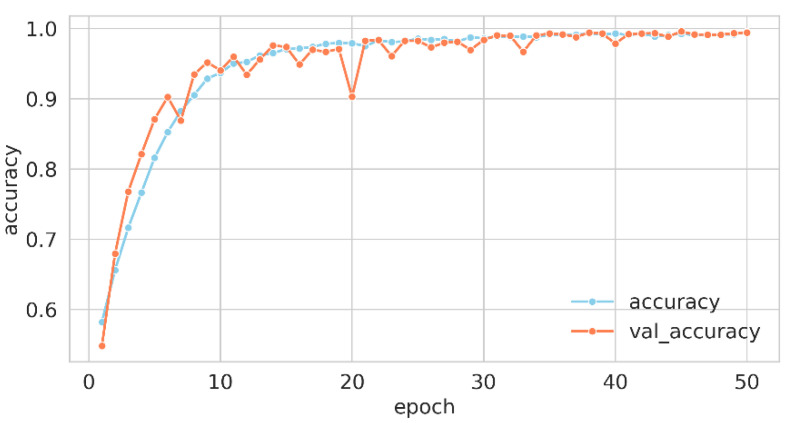
Training accuracy curve for proposed net.

**Figure 4 children-10-01654-f004:**
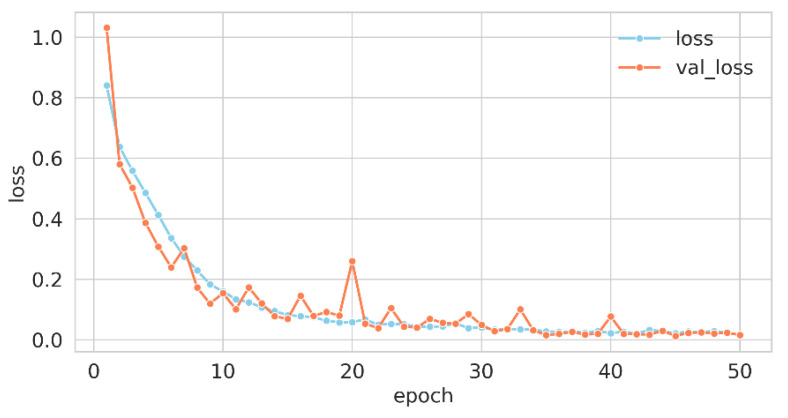
Training loss curve for proposed net.

**Figure 5 children-10-01654-f005:**
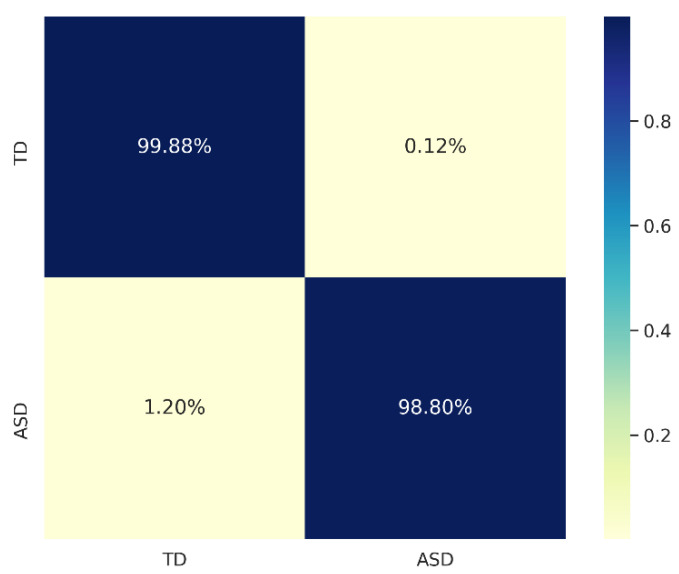
Confusion matrix for ASD and TD classification.

**Figure 6 children-10-01654-f006:**
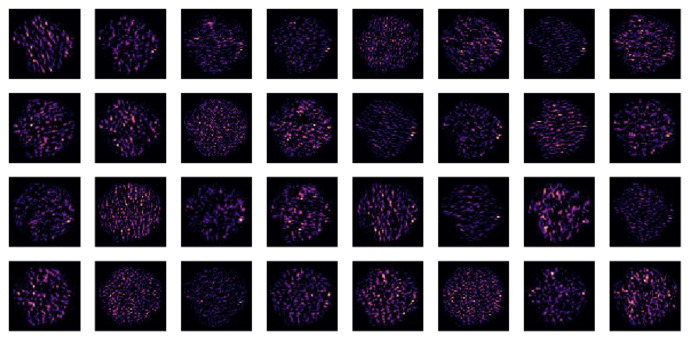
The feature map of conv2d 0 layer.

**Figure 7 children-10-01654-f007:**
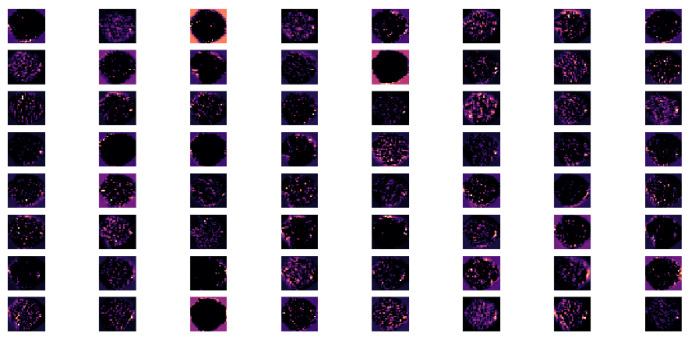
The feature map of conv2d 1 layer.

**Figure 8 children-10-01654-f008:**
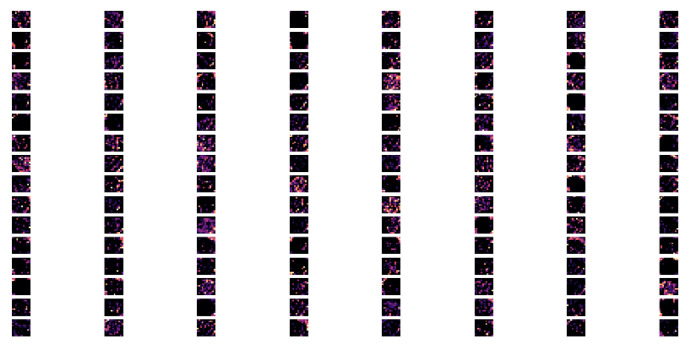
The feature map of conv2d 2 layer.

**Figure 9 children-10-01654-f009:**
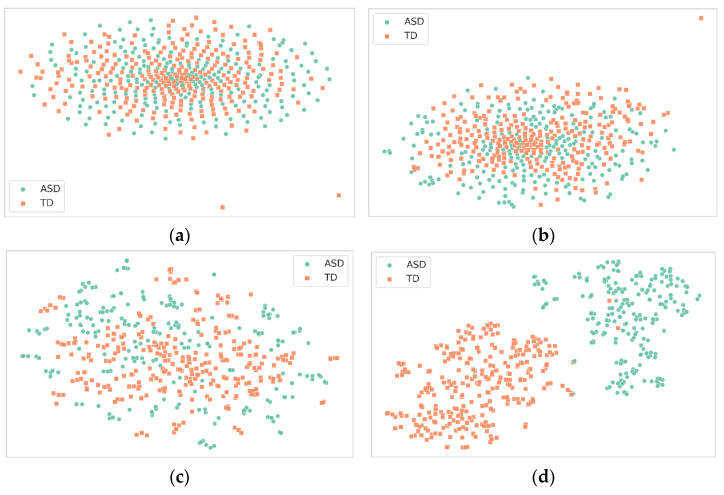
Visualizing t-SNE-reduced outputs of the proposed model’s layers: (**a**) first convolutional layer, (**b**) second convolutional layer, (**c**) third convolutional layer, (**d**) initial fully connected layer.

**Table 1 children-10-01654-t001:** The phenotypic characteristics of children who participated in this study.

Category	Value
Total	126
ASDs	56
TDs	70
ASD Males	51
ASD Females	5
TD Males	51
TD Females	19
Age Range of ASDs (Years)	7.13–18.00
Age Range of TDs (Years)	6.47–18.00
Average Age (SD) (Years)	12.02 (2.97)
Average ADOS scores of ASDs (SD)	11.20 (4.27)

**Table 2 children-10-01654-t002:** Configuration and parameterization of layers in the proposed network architecture.

Layer (Type)	Output Shape	Param
conv2d 0	(None, 62, 62, 32)	896
batch normalization	(None, 62, 62, 32)	128
max pooling2d	(None, 31, 31, 32)	0
conv2d 1	(None, 29, 29, 64)	18,496
batch normalization 1	(None, 29, 29, 64)	256
max pooling2d 1	(None, 14, 14, 64)	0
conv2d 2	(None, 12, 12, 128)	73,856
batch normalization 2	(None, 12, 12, 128)	512
max pooling2d 2	(None, 6, 6, 128)	0
flatten	(None, 4608)	0
dropout	(None, 4608)	0
dense	(None, 1024)	4,719,616
batch normalization 3	(None, 1024)	4096
dropout 1 (Dropout)	(None, 1024)	0
dense 1 (Dense)	(None, 256)	262,400
batch normalization 4	(None, 256)	1024
dropout 2 (Dropout)	(None, 256)	0
dense 2 (Dense)	(None, 64)	16,448
batch normalization 5	(None, 64)	256
dropout 3 (Dropout)	(None, 64)	0
dense 3 (Dense)	(None, 2)	130

**Table 3 children-10-01654-t003:** Evaluation of the proposed model’s performance on the test dataset.

Metric	Accuracy (%)	Precision (%)	Recall (%)	F1-Score (%)
Value	99.39	99.85	98.80	99.32

**Table 4 children-10-01654-t004:** Accuracies of five-fold cross-validation.

Fold Number	1	2	3	4	5
Accuracy (%)	99.39	99.68	99.70	99.32	99.09

**Table 5 children-10-01654-t005:** ASD classification accuracy across various models.

References	Method	Pattern	Purpose	Accuracy (%)
Yakolli et al. [38]	CNN	FC and structural	Classification	88.0
Jiang et al. [39]	3D CNN	FC	Classification	72.46
Hao et al. [12]	Deep-broad learning	ROI	Classification	71.8
Husna et al. [40]	CNN	FC	Classification	87.0
Shi et al. [41]	Decision model	ROI	Classification	75.41
Niu et al. [42]	DANN	FC/ROI	Classification	73.2
Byeon et al. [16]	RNN	FC	Classification	74.54
Jiao et al. [15]	CapsNets	FC	Classification	71.0
Yin et al. [14]	DNN, AE	ROI	Classification	79.20
Anirudh et al. [13]	GCNN	ROI	Classification	70.86
Zhao et al. [11]	3D CNN	ROI	Classification	70.1
Guo et al. [43]	DNN	FC	Classification	86.36
Ktena et al. [44]	GCNN	ROI	Classification	62.9
The proposed model	CNN	FC	Classification	99.39

## Data Availability

We thank the numerous contributors to the ABIDE database for their efforts in the collection, organization, and sharing of their datasets. The data that support the findings of this study are openly available in the ABIDE database at http://fcon_1000.projects.nitrc.org/indi/abide/abide_I.html (accessed on 27 September 2023).

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
