# Peer review of "Detection of ASD Children through Deep-Learning Application of fMRI"

_children, 2023, doi:10.3390/children10101654_

Round 1

Reviewer 1 Report

Title: Detection of ASD Children through Deep Learning Analysis of fMRI

The manuscript delves into a pivotal subject, addressing the detection of Autism Spectrum Disorder (ASD) utilizing neuroimaging data and deep learning algorithms. While the accuracy achieved in identifying ASD children holds promise, several refinements are essential to enhance the manuscript's overall quality.

Introduction:

1.       To bolster the introduction, it is advisable to encompass a broader discussion of fMRI findings in both children with and without ASD. This could include highlighting notable discoveries such as increased local connectivity and decreased long-range connectivity, thereby substantiating the selection of fMRI variables for integration into the deep learning algorithm. Additionally, a succinct summary of previous deep learning studies, encompassing the fMRI data employed and the characteristics of the participating children, should be integrated.

2.       The content currently found in lines 58 to 65, which outlines the findings and their conclusions, might be better situated in later sections, specifically after the elucidation of methodologies and experimental paradigms.

Method:

1.       In Table 1, it is recommended to present age and gender data separately for the TD and ASD groups. Furthermore, the inclusion of statistical analyses to ascertain the equivalence of age and gender distribution between these groups is essential.

2.       To enhance comprehensibility, provide more comprehensive details regarding fMRI data acquisition. For instance, specify whether resting state data was collected under open-eye or closed-eye conditions, the duration of the resting-state period, and whether the exclusion rate remained consistent across the TD and ASD groups.

3.       Augment the description of fMRI data processing and elaborate on the specific variables incorporated into the algorithm.

Discussion:

1.       Expanding the current discussion to include insights into how unique architectural elements and feature extraction strategies contribute to the superior detection accuracy of the current model in comparison to prior models is advised. Exploring potential other elements that may influence these differences, such as children's age, ASD severity, or choice of fMRI variables, would further enrich the discussion.

2.       Given that the present study concentrates on school-age children, the applicability of its findings to younger populations remains uncertain. This poses a potential challenge in applying the model for early identification of ASD in children. To address this, it is suggested to include a more comprehensive discussion on this point and potentially acknowledge it as a limitation.

3.       Considering the challenges associated with acquiring fMRI data from children with ASD due to sensory sensitivities and behavioral issues, it may be worthwhile to explore alternative, child-friendly neuroimaging tools like functional near-infrared spectroscopy (fNIRS) and EEG. Acknowledging this potential limitation and suggesting avenues for future research in this direction would be a valuable addition.

Author Response

Thank you for your thorough and constructive review of our manuscript titled "Detection of ASD Children through Deep Learning Analysis of fMRI." Your insights have been invaluable in enhancing the quality and rigor of our work. The modified sections are highlighted in the revised version.

Title: Detection of ASD Children through Deep Learning Analysis of fMRI

The manuscript delves into a pivotal subject, addressing the detection of Autism Spectrum Disorder (ASD) utilizing neuroimaging data and deep learning algorithms. While the accuracy achieved in identifying ASD children holds promise, several refinements are essential to enhance the manuscript's overall quality.

Response:

Thank you for recognizing the importance of our work in the field of Autism Spectrum Disorder(ASD) detection using neuroimaging data and deep learning algorithms.

Introduction:

  1. To bolster the introduction, it is advisable to encompass a broader discussion of fMRI findings in both children with and without ASD. This could include highlighting notable discoveries such as increased local connectivity and decreased long-range connectivity, thereby substantiating the selection of fMRI variables for integration into the deep learning algorithm. Additionally, a succinct summary of previous deep learning studies, encompassing the fMRI data employed and the characteristics of the participating children, should be integrated.

Response: Thank you for your insightful comment. we have expanded the introduction to include specific fMRI findings in children with and without ASD, such as increased local and decreased long-range connectivity, and also added a brief summary of previous deep learning studies that have employed fMRI data.

  1. The content currently found in lines 58 to 65, which outlines the findings and their conclusions, might be better situated in later sections, specifically after the elucidation of methodologies and experimental paradigms.

Response: Thank you for your suggestion; we have relocated the content from lines 58 to 65 to Section 4.3, under the "Discussion" heading, to better align with the presentation of methodologies and experimental paradigms.

Method:

  1. In Table 1, it is recommended to present age and gender data separately for the TD and ASD groups. Furthermore, the inclusion of statistical analyses to ascertain the equivalence of age and gender distribution between these groups is essential.

Response: We appreciate the recommendation and will update Table 1 to display age and gender data distinctly for both the TD and ASD groups, as detailed in the revised Table 1.

  1. To enhance comprehensibility, provide more comprehensive details regarding fMRI data acquisition. For instance, specify whether resting state data was collected under open-eye or closed-eye conditions, the duration of the resting-state period, and whether the exclusion rate remained consistent across the TD and ASD groups.

Response:

We have expanded the section on fMRI data acquisition to include specific details such as the eye conditions during resting-state scans, the duration of the resting-state period.

  1. Augment the description of fMRI data processing and elaborate on the specific variables incorporated into the algorithm.

Response: We have enriched Section 2.3, Paragraph 1, to provide a comprehensive overview of fMRI data preprocessing and to delineate the specific variables employed in the optimization of the CNN algorithm for superior ASD classification.

Discussion:

  1. Expanding the current discussion to include insights into how unique architectural elements and feature extraction strategies contribute to the superior detection accuracy of the current model in comparison to prior models is advised. Exploring potential other elements that may influence these differences, such as children's age, ASD severity, or choice of fMRI variables, would further enrich the discussion.

Response: Thank you for your insightful recommendation. In the discussion section, we have enriched the analysis by detailing how the model's distinct architectural components and feature extraction methodologies enhance its detection accuracy, while also examining additional influencing factors such as pediatric age range, ASD severity, and fMRI variable selection.

  1. Given that the present study concentrates on school-age children, the applicability of its findings to younger populations remains uncertain. This poses a potential challenge in applying the model for early identification of ASD in children. To address this, it is suggested to include a more comprehensive discussion on this point and potentially acknowledge it as a limitation.

Response: We appreciate the insightful comment and agree that the study's focus on school-age children limits its applicability to younger populations; this limitation will be explicitly acknowledged and discussed more comprehensively in the section dedicated to limitations.

  1. Considering the challenges associated with acquiring fMRI data from children with ASD due to sensory sensitivities and behavioral issues, it may be worthwhile to explore alternative, child-friendly neuroimaging tools like functional near-infrared spectroscopy (fNIRS) and EEG. Acknowledging this potential limitation and suggesting avenues for future research in this direction would be a valuable addition.

Response: We appreciate the insightful suggestion regarding the challenges of fMRI data acquisition in children with ASD and agree that exploring alternative, child-friendly neuroimaging tools like fNIRS and EEG could offer valuable avenues for future research; we will duly acknowledge this limitation in the revised manuscript.

Your feedback has been instrumental in refining our manuscript, and we are grateful for the time and expertise you have invested in this review.

Reviewer 2 Report

In the present study, a deep learning algorithm called “Convolutional Neural Network (CNN) was demonstrated to have an extraordinary accuracy rate ex exceeding 99% in the classification of ASD. For the study authors used fMRI datasets of ABIDE. The authors propose that this algorithm can be used as an early diagnostic tool for ASD. Limitations of the method in assessing severity and subtyping are also discussed. My comments are minor points to bring the manuscript to the next level.    Major revisions    Comment 1 The source of data and respective sample size of the cases and controls need to be mentioned in the abstract.    Minor revisions    Comment 1 The first sentence of the introduction is incomplete. It overlooks essential criterion B for the diagnosis of ASD in DSM-5. “Restricted, repetitive patterns of behavior, interests, or activities”. This could mislead naïve readers., It is therefore advisable to add criterion B as well.    Comment 2  Define abbreviations at the first mention and use them consistently thereafter. e.g. Define FLIRT, and AFNI (what they stand for) at itheir first mention. Line 156- Convolutional Neural Network (CNN)      Comment 3 It would be good to highlight other benefits/ applications of fMRI·      fMRI is pertinent in personalized metabolic therapies in ASD. E.g. integrated PET/MRI system which is useful in ASD.  ·      No exposure to radiation.    Comment 4 Mention briefly the observations/interpretations of Figure 6 to 8, in the respective figure captions.      Comment 5  Acknowledgment: As per the Usage Agreement http://fcon_1000.projects.nitrc.org/indi/abide/abide_I.html  , the specific datasets included in analyses should be characterized and specified appropriately, and their funding sources should be acknowledged. 

Consult the following paper.

Nielsen JA, Zielinski BA, Fletcher PT, Alexander AL, Lange N, Bigler ED, Lainhart JE, Anderson JS. Multisite functional connectivity MRI classification of autism: ABIDE results. Front Hum Neurosci. 2013 Sep 25;7:599. doi: 10.3389/fnhum.2013.00599. PMID: 24093016; PMCID: PMC3782703.

Comment 6

Compare the findings of the present study with the previous studies that used a similar approach.

Jiang W, Liu S, Zhang H, Sun X, Wang SH, Zhao J, Yan J. CNNG: A Convolutional Neural Networks With Gated Recurrent Units for Autism Spectrum Disorder Classification. Front Aging Neurosci. 2022 Jul 5;14:948704. doi: 10.3389/fnagi.2022.948704. PMID: 35865746; PMCID: PMC9294312.

Author Response

Thank you for your comprehensive review and constructive feedback, which we find invaluable for enhancing the quality of our manuscript. The modified sections are highlighted in the revised version.

In the present study, a deep learning algorithm called “Convolutional Neural Network (CNN)” was demonstrated to have an extraordinary accuracy rate ex exceeding 99% in the classification of ASD. For the study authors used fMRI datasets of ABIDE. The authors propose that this algorithm can be used as an early diagnostic tool for ASD. Limitations of the method in assessing severity and subtyping are also discussed. My comments are minor points to bring the manuscript to the next level.   

Response: Thank you for your thoughtful review and for recognizing the potential of our work. Your comments are indeed valuable for elevating the quality of our manuscript to the next level.

Major revisions   

Comment 1: The source of data and respective sample size of the cases and controls need to be mentioned in the abstract.   

Response: We agree that the source of data and respective sample sizes for cases and controls should be mentioned in the abstract. This has been added for clarity.

Minor revisions   

Comment 1: The first sentence of the introduction is incomplete. It overlooks essential criterion B for the diagnosis of ASD in DSM-5. “Restricted, repetitive patterns of behavior, interests, or activities”. This could mislead naïve readers., It is therefore advisable to add criterion B as well.   

Response: We appreciate your attention to detail. The first sentence of the introduction has been revised to include criterion B for the diagnosis of ASD in DSM-5, to avoid misleading readers.

Comment 2:  Define abbreviations at the first mention and use them consistently thereafter. e.g. Define FLIRT, and AFNI (what they stand for) at itheir first mention. Line 156- Convolutional Neural Network (CNN)

 Response: Abbreviations like FLIRT and AFNI have been defined at their first mention, as you suggested, to ensure consistency throughout the manuscript.

Comment 3: It would be good to highlight other benefits/ applications of fMRI·      fMRI is pertinent in personalized metabolic therapies in ASD. E.g. integrated PET/MRI system which is useful in ASD.  ·      No exposure to radiation.  

Response: We have elaborated on the additional advantages of fMRI in Section 1 of the Introduction, emphasizing its role in personalized metabolic therapies for ASD and its absence of radiation exposure.

 Comment 4: Mention briefly the observations/interpretations of Figure 6 to 8, in the respective figure captions.  

Response: The captions for Figures 6 to 8 have been succinctly annotated to enhance comprehension.

Comment 5: Acknowledgment: As per the Usage Agreement http://fcon_1000.projects.nitrc.org/indi/abide/abide_I.html  , the specific datasets included in analyses should be characterized and specified appropriately, and their funding sources should be acknowledged.

Consult the following paper.

Nielsen JA, Zielinski BA, Fletcher PT, Alexander AL, Lange N, Bigler ED, Lainhart JE, Anderson JS. Multisite functional connectivity MRI classification of autism: ABIDE results. Front Hum Neurosci. 2013 Sep 25;7:599. doi: 10.3389/fnhum.2013.00599. PMID: 24093016; PMCID: PMC3782703.

Response: We appreciate the reviewer's guidance on adhering to the Usage Agreement for the ABIDE dataset. In response, we will revise the Data Availability Statement section to specify the datasets used, characterize them appropriately, and acknowledge their funding sources, while also consulting the recommended paper for further context.

Comment 6

Compare the findings of the present study with the previous studies that used a similar approach.

Jiang W, Liu S, Zhang H, Sun X, Wang SH, Zhao J, Yan J. CNNG: A Convolutional Neural Networks With Gated Recurrent Units for Autism Spectrum Disorder Classification. Front Aging Neurosci. 2022 Jul 5;14:948704. doi: 10.3389/fnagi.2022.948704. PMID: 35865746; PMCID: PMC9294312.

Response: Thank you for pointing out the need for a comparative analysis with previous studies. We have indeed included such a comparison in Table 4 and cited the relevant study by Jiang et al. for a comprehensive understanding.

Your comments have been instrumental in refining our manuscript, and we are grateful for the expertise and time you have invested in this review.

Reviewer 3 Report

Interesting work, I would like to thank the authors for this contribution. The study presents an interesting ML application of in the autism context. However, please consider the points below in the next version.

(1)

The dataset used by the study has been available since 2014. That said, there is a need for additional clarification on the specific motivations of the research. For example, does literature generally lack studies that kind of models?

(2)

I am wondering if any precautions have been taken to ensure that the test set did not contain samples associated with subjects present in the training set. This precaution is of paramount importance to prevent any potential information leakage, which could erroneously inflate performance metrics. My concerns in this regard are particularly heightened, given the model's remarkable achievement of nearly 100% performance.

(3)

Given the limited size of the dataset (only 126 subjects), it is imperative to validate the model through cross-validation. This would help mitigate the risk of overfitting and provides a more robust assessment of the model's performance.

(4)

The literature review demonstrates a commendable effort. However, it would be valuable to expand the discussion on eye-tracking methods coupled with ML for autism diagnosis. Eye-tracking is particularly one of the widely used approaches in the context of autism diagnosis. One possible example, applying Deep Learning, to mention:

https://doi.org/10.2196/27706

(5)

Multiple references should be cited, please. This includes the references of Adam, Batch Normalization, and Dropout.

(6)

Please discuss possible ethical considerations related to the potential implications of using AI techniques for diagnosing ASD. Addressing ethical concerns and potential biases in the models is important for responsible research.

Overall, I appreciate the authors' efforts and look forward to seeing an improved version of this study.

The quality of language is generally good.

Author Response

Thank you for your thoughtful comments and constructive feedback on our manuscript. We appreciate the time you've invested in reviewing our work. The modified sections are highlighted in the revised version.

Interesting work, I would like to thank the authors for this contribution. The study presents an interesting ML application of in the autism context. However, please consider the points below in the next version.

Response:

Thank you for your positive remarks and for recognizing the contribution of our study in the context of autism and machine learning.

(1) The dataset used by the study has been available since 2014. That said, there is a need for additional clarification on the specific motivations of the research. For example, does literature generally lack studies that kind of models?

Response:  Thank you for your insightful comment. We have provided further clarification on the specific motivations behind our research.

(2) I am wondering if any precautions have been taken to ensure that the test set did not contain samples associated with subjects present in the training set. This precaution is of paramount importance to prevent any potential information leakage, which could erroneously inflate performance metrics. My concerns in this regard are particularly heightened, given the model's remarkable achievement of nearly 100% performance.

Response: Thank you for your astute feedback. To mitigate the risk of information leakage and ensure the integrity of our performance metrics, we partitioned the data at the subject level for the training, validation, and test sets. Further insights into the model's exceptional performance are elaborated upon in the discussion section.

(3) Given the limited size of the dataset (only 126 subjects), it is imperative to validate the model through cross-validation. This would help mitigate the risk of overfitting and provides a more robust assessment of the model's performance.

Response: Thank you for your insightful suggestion. I have incorporated cross-validation into the analysis to address the concern raised about the dataset. This ensures a more robust evaluation of the model's performance.

(4) The literature review demonstrates a commendable effort. However, it would be valuable to expand the discussion on eye-tracking methods coupled with ML for autism diagnosis. Eye-tracking is particularly one of the widely used approaches in the context of autism diagnosis. One possible example, applying Deep Learning, to mention:

https://doi.org/10.2196/27706

Response: Thank you for your constructive feedback on the literature review. We acknowledge the significance of eye-tracking methods in autism diagnosis and will expand our discussion to include its integration with machine learning techniques, referencing the example you provided.

(5) Multiple references should be cited, please. This includes the references of Adam, Batch Normalization, and Dropout.

Response: Thank you for your valuable suggestion. We have updated the manuscript to include multiple references, specifically for Adam optimization, Batch Normalization, and Dropout techniques.

(6) Please discuss possible ethical considerations related to the potential implications of using AI techniques for diagnosing ASD. Addressing ethical concerns and potential biases in the models is important for responsible research.

Response: Thank you for emphasizing the necessity of discussing ethical considerations in AI applications for ASD diagnosis. We have added a section to the final paragraph of the discussion to explicitly address these ethical concerns.

Overall, I appreciate the authors' efforts and look forward to seeing an improved version of this study.

Response: Thank you again for your constructive feedback and for recognizing the efforts invested in this study.

Round 2

Reviewer 3 Report

Thanks for accommodating the feedback. I have no further comments.